# Optimising Speaker-Dependent Feature Extraction Parameters to Improve Automatic Speech Recognition Performance for People with Dysarthria

**DOI:** 10.3390/s21196460

**Published:** 2021-09-27

**Authors:** Marco Marini, Nicola Vanello, Luca Fanucci

**Affiliations:** Department of Information Engineering, University of Pisa, Via G. Caruso 16, 56122 Pisa, Italy; nicola.vanello@unipi.it (N.V.); luca.fanucci@unipi.it (L.F.)

**Keywords:** dysarthria, automatic speech recognition, speech analysis, genetic algorithm, kaldi

## Abstract

Within the field of Automatic Speech Recognition (ASR) systems, facing impaired speech is a big challenge because standard approaches are ineffective in the presence of dysarthria. The first aim of our work is to confirm the effectiveness of a new speech analysis technique for speakers with dysarthria. This new approach exploits the fine-tuning of the size and shift parameters of the spectral analysis window used to compute the initial short-time Fourier transform, to improve the performance of a speaker-dependent ASR system. The second aim is to define if there exists a correlation among the speaker’s voice features and the optimal window and shift parameters that minimises the error of an ASR system, for that specific speaker. For our experiments, we used both impaired and unimpaired Italian speech. Specifically, we used 30 speakers with dysarthria from the IDEA database and 10 professional speakers from the CLIPS database. Both databases are freely available. The results confirm that, if a standard ASR system performs poorly with a speaker with dysarthria, it can be improved by using the new speech analysis. Otherwise, the new approach is ineffective in cases of unimpaired and low impaired speech. Furthermore, there exists a correlation between some speaker’s voice features and their optimal parameters.

## 1. Introduction

Dysarthria is a motor speech disorder caused by a neurological injury [1] affecting the brain areas responsible for speech. This damage manifests itself in different ways in different subjects, leading to several types of dysarthria. Speech disorders can affect the production, rhythm, pitch, rate, loudness, quality, and duration of speech. Dysarthric people comprise individuals with a primary speech disorder or those who experience a speech disorder as a result of a disease such as amyotrophic lateral sclerosis (ALS) or Parkinson’s Disease (PD). Dysarthria reduces speech intelligibility, thus affecting social interaction and quality of life.

Despite the fact that in the last several years Automatic Speech Recognition (ASR) systems have recorded an outstanding performance improvement, current commercial technology cannot offer a good solution to help people affected by speech disabilities [2]. This is a crucial point because people affected by speech impairments usually also have other kinds of disabilities. For instance, people affected by Parkinson’s also have mobility impairments so, a Virtual Assistant (VA) with a hands-free interface that can understand their vocal commands can really help people with Parkinson’s in their daily lives.

For this reason, the most important challenge is to build an ASR system able to understand a dysarthric utterance. The main implementations of such system are based on the Hidden Markov Model (HMM) combined with the Gaussian Mixture Model (GMM) [3] or on a Deep Neural Network (DNN) [4].

Nowadays, ASR systems and hands-free interfaces are not designed to process commands from a person with dysarthria, because they are mainly trained on unimpaired speech. In fact, the pronunciation of speakers with dysarthria deviates from that of non-disabled speakers in many aspects. Thus, state-of-the-art approaches in ASR show very low performances when applied to dysarthric speech [5].

In order to design an ASR system optimised for dysarthric speech, a large amount of dysarthric data is needed. In the last years, the interest of the scientific community in this topic has grown. As a result, several dysarthric speech databases have been created, especially in English [6,7,8]. Moreover, preliminary studies on ASR systems for dysarthric speech have been presented. An earlier study about acoustic and lexical model adaptation was performed in [9]. This study showed a Word Error Rate (WER) average reduction of 36.99% for an ASR system trained over a large vocabulary dysarthric speech database.

Another interesting approach comes from the idea of tuning GMM-HMM parameters of ASR systems that have been developed for unimpaired speech. In [10], TORGO database [6] was exploited to perform a Dysarthric speech recognition task. An acoustic model for a dysarthric speech recognition system using GMM-HMMs and DNN-HMMs was adopted. This approach focused on the tuning of speaker-specific parameters. A relative WER reduction of 17.62% with respect to the baseline system trained on a more complex model was reported. A comparative study among different architectures was performed in [11]. The results show that hybrid DNN-HMM models outperform classical GMM-HMM ones according to WER measures. The database used was TORGO database [6] and a 13% improvement in WER was achieved with respect to the classical architectures.

ASR systems are based on the estimation of speech features using a moving window approach. Specifically, for each utterance, a feature matrix vector is created. This matrix comprises different speech features, such as Mel-Feature Cepstral Coefficient (MFCC) or Perceptual Linear Prediction (PLP) [12], that are estimated in different time intervals, i.e. time windows, that span the word or sentence to be recognised.

Most ASR systems use a fixed frameshift and window size in speech processing. This is based on the assumption that the non-stationary speech signal can be approximated by a piecewise quasi-stationary process. The common choice of 25 ms window and 10 ms shift size is a compromise between data rate and resolution determined empirically to give a reasonable performance on average. In ASR systems for dysarthric speech, these values might not represent the optimal choice. All the mentioned studies on ASR systems for dysarthric speech [9,10,11] share the same values for these parameters. Specifically, they adopt a value of 15 ms as a time step in the moving window procedure instead of 10 ms as included in the standard approach. The window size is 25 ms that is the value usually adopted in ASR for unimpaired speech.

Starting from the above-mentioned findings, in [13] window and shift sizes were optimised at a single-subject level, thus adopting a speaker-dependent (SD) approach. Specifically, the goal was to estimate subject-specific values for such parameters able to minimize the WER. The ASR system was developed by using Kaldi toolkit [14] and it was trained over an Italian dysarthric database composed of 5 speakers. The results showed that there exists an Optimal Region (OR) in the window and shift field where the ASR performance is optimised with respect to the standard values. The observed improvements in the WER ranged from 31% to 81% according to the selected speaker.

Although promising, these results were obtained using a small Italian speakers with dysarthria database that contains only five speakers (three males and two females). Furthermore, no comparison of developed ASR performance was performed by using unimpaired speech samples.

This work represents an extension of the paper presented in the proceedings of the 2020 International Conference on Applications in Electronics Pervading Industry, Environment and Society. In this work, we aim at significantly improving the work in [13] by using a bigger Italian dysarthric database as well as an Italian database of unimpaired speech. The main purpose of this work is to confirm the preliminary findings described in [13] and to further critically explore the results by looking in at the possible role of subject-specific speech features in determining the time window parameters. To achieve this goal, a relationship between the speaker’s OR and speech features will be introduced.

The findings of this work could be useful to fine-tune the ASR system according to each speaker characteristics, thus optimising the ASR system performance without recording a huge amount of hours of the speaker’s voice.

We describe the materials and methods used in Section 2 before discussion of the experimental setups in Section 3. In Section 4 we report the experimental results and then in Section 5 we approach a discussion about them. Finally, conclusions are drawn in Section 6.

## 2. Materials and Methods

### 2.1. IDEA: The Dysarthric Database

As said in the previous section, in the last years a lot of effort has been spent to collect dysarthric voices. Very recently, in [15] the authors collected the voices of 45 speakers with dysarthria (20 females and 25 males) affected by 8 different pathologies in the IDEA database. The total amount of recordings is 16794, resulting in 13.72 h of recording. The records consist of 211 isolated words repeated at most three times by each speaker. The IDEA database is not balanced, which means not all speakers have the same amount of records. Furthermore, an additional annotation procedure has been done in order to clean up the database from corrupted or not usable records. This process has reduced the total amount of usable records by 6.4%. Only 70.8% of records can be used as it is (the annotation TextGrid file does not present any note), the other 22.8% present some issues such as background noise, word truncation, etc.

### 2.2. Speaker Selection

In this work, we decided to take into account just the highest quality recordings, that is 70.8% of the total records. For each speaker, we generated the training and test sets as explained in [15]. By training and test sets we mean a set of recordings that will be used to train and test the system. Since the database is unbalanced, training and test sets are unbalanced as well. Table 1 shows the records contained in the training and test sets for each speaker, sorted by the number of records contained in the test set.

The experiment in [13] that we want to replicate consisted of finding out the best couple of values of window and shift parameters that optimise the performance of an ASR system trained with a speaker-dependent (SD) approach. The SD approach means: for the training and test processes, we have to use the record of a single speaker. Therefore, all the speakers that have empty test sets must be ruled out. Furthermore, speakers who have just a few records in the test set generate WER not comparable with other speakers, so we decided to use just the speakers who have the number of records in the test set greater than 20. Thus, our dataset is composed of the first 30 speakers listed in Table 1.

### 2.3. CLIPS: Unimpaired Speech Database

The CLIPS database http://www.clips.unina.it/it/index.jsp (accessed on 1 September 2020) is a publicly available corpus of Italian speech. It collects different type of corpora that are divided into: ‘ortofonico’ (that means speech therapy), ‘telefonico’ (telephone dialogue), ‘letto’ (readings), ‘dialogico’ (spontaneous speech), and ’radiotelevisivo’ (radio and TV programs). The ‘telefonico’ is composed by raw audio files with a sampling rate of 8000 Hz, while ‘ortofonico’ wave files were sampled at 44,100 Hz. All other corpora were sampled at 22,050 Hz.

For our experiment, we decided to use the ‘ortofonico’ corpora because it contains the recordings of 10 professional speakers (5 females and 5 males) and all the recordings are transcribed and annotated. Each speaker carries out 3 repetitions of 20 utterances (LP list) suitable for lexical coverage and a single repetition of 120 utterances (LB list) suitable for completing phonemes and phonetic links. The amount of vocal material available can be estimated at around 14 minutes for each speaker, which means a total of about 2.5 h for the whole ‘ortofonico’ corpus.

For our experiment with the CLIPS database, we decided to use all the 120 utterances of the LB list and 2 of 3 repetitions of the LP list for the training set, while just 1 repetition of the LP list for the test set. Therefore, for each speaker, we have 160 recordings (120 records from LB and 40 records from LP) for the training set and 20 recordings from LP for the test set. To test the ASR system we chose recordings that were not used in the training phase.

### 2.4. Automatic Speech Recognition System and Genetic Algorithm Implementation

As said in the previous section, we decided to use a SD approach because we want to find out what are the best parameters at the feature extraction level, that optimise the performance of an ASR system according to the speaker selected. The ASR architecture is based on a Gaussian Mixture Model (GMM) combined with Hidden Markov Model (HMM) [16]. For our purpose, we decided a tri-phoneme Acoustic Model trained with Speaker Adaptive Training (SAT) algorithm. The features vector is projected by Linear Discriminant Analysis (LDA) criterion [17] and transformed by Maximum Likelihood Linear Transformation (MLLT) [18] (***LDA**+**MLLT**+**SAT***) trained with records of a single user. We used the Kaldi toolkit [14] to develop the ASR system following the recipes https://github.com/marcomarini93/idea.git (accessed on 1 March 2021) explained in [15].

The genetic algorithm (GA) was developed using the Python library https://pypi.org/project/geneticalgorithm2/ (accessed on 20 April 2021).

GAs are adaptive heuristic search algorithms based on the evolutionary ideas of natural selection and genetics. As such they represent an intelligent exploitation of a random search used to solve optimization problems. GA simulates the survival of the fittest among individuals over consecutive generation for solving a problem. Each generation consists of a population of character strings that are analogous to the chromosome that we see in our DNA. Each individual represents a point in a search space and a possible solution. The individuals in the population are then made to go through a process of evolution that is summarised by Algorithm 1.
**Algorithm 1** Genetic Algorithm process steps1:Randomly initialize population (t)2:Determine fitness of population (t)3:**repeat**4:    Select parents from population (t)5:    Perform crossover on parents creating population (t + 1)6:    Perform mutation of population (t + 1)7:    Determine fitness of population (t + 1)8:**until** best individual is good enough

In our scenario, each individual represents a point in *Window-Shift* field, and we use the WER measure of the ASR system as *Fitness Function*. So, in the fitness function, the Python program first modifies the Kaldi configuration file through its *Window-Shift* values and then executes the Kaldi recipe which returns a WER value. To summarise, the elements of GA are:Chromosome: composed by 2 genes (float values) that represent Window and Shift parameters;Crossover: single point;Likelihood Mutation: 0.15;Population Size: 50;Stop Criterion: Fitness convergence ($10^{-4}$);

## 3. Experimental Setup

Two experiments were planned. The first one aims at exploring the possibility of a subject-specific optimal region for moving window size and shift. Specifically, we will estimate whether an OR for the above-mentioned parameters exists, such that ASR performance is maximised. The validity of the proposed approach will be verified by looking at the optimal region estimated on the unimpaired speech dataset. The second experiment will aim at finding possible correlations among relevant speech features and OR. This would allow estimating the optimal region on a subject-by-subject basis, minimising computation time and training data acquisition.

### 3.1. First Experiment

As said in Section 2.2, for the first experiment, we decided to use just 30 IDEA speakers. Figure 1 shows the general experiment process. It consists of the selection of a speaker and generates the training and test sets from his recordings. Then, the speech features based on the Window and Shift sizes selected by the Genetic Algorithm are estimated.

With these features, in particular, the MFCC [12], we train and test an ASR system generating a Word Error Rate (WER) value that measures the performance of the ASR system. The GA will use the WER value in order to generate a new couple of Window and Shift sizes that should minimize the next WER. The last step is repeated until the GA achieves convergence conditions. The optimal region for each speaker is defined using the mean and standard deviation of the window size parameter, and the mean and standard deviation of the window shift parameter.

Algorithm 2 describes the experiment algorithm, implemented in python. The Feature Extraction and ASR blocks are implemented in the bash script (Kaldi recipe).
**Algorithm 2** Experiment process steps1:Select a speaker2:Generate Training and Test sets3:**repeat**4:    Feature Extraction by using Window and Shift size given by GA5:    Train and Test ASR system6:    Compute ASR WER7:    Change Window and Shift size based on WER8:**until** GA converges9:Compute the speaker’s Optimal Region vector

### 3.2. Second Experiment

The second experiment aims to find possible correlations between speakers characteristics, as captured by speech features both in the time and frequency domains, and OR vector parameters.

Figure 2 shows the second experiment’s procedure. As a result of the first experiment, an OR vector for each speaker is obtained. A set of 24 supra-segmental speech features are estimated for each word. To estimate speech features, 15 words that are common across all subjects are selected. The speech feature vector for each subject is estimated by computing the mean of each feature obtained from each word. Then, a correlation analysis has been done among Window and Shift mean values of OR vector and 24 speech features. In the correlation analysis, we compute the correlation coefficients and the relative *p*-values. A Benjamini–Yekuteli (BY) correction was performed to control for the false discovery rate in this multiple comparison analysis. Specifically, a false discovery rate equal to 0.1 was set. The BY procedure was performed using the more conservative hypothesis about generic data covariance structure. i.e. linear dependency among the different tests [19].

The 15 words used for speaker’s features extraction are:accendicancellochiesacorridoiofannoforzahannomaggiorepaesipubblicoscenariosoffittasvegliaterrazzozero

About the supra-segmental speech features, we selected some speech features, describing glottal source, voice quality, timing information, prosody, as well as speech frequency content. Some of those are used for assessment of voice quality [20]. Specifically, we took into account classical features such as duration of the file in seconds, fundamental frequency F0 and its standard deviation, Subharmonics and Harmonics information, Jitter [21], Shimmer [22], Formants information, Intensity, Speech Rate, Signal to Noise Ratio, Long-Term average spectrum (LTAS) [23], Root Mean Square energy (RMS), and some spectral shape information. In particular, the list of the 24 features extracted is:**Duration in seconds**: it is the length of the voice file in seconds;**Mean F0**: it is the mean of the Fundamental frequency F0;**STD F0**: it is the standard deviation of the Fundamental frequency F0;**Subharmonics to Harmonic ratio (SHR)**: amplitude ratio between subharmonics and harmonics according to [24];**Subharmonics pitch**: the fundamental frequency estimate introduced in [24] for impaired speakers;**Local jitter**: parameter of frequency variation from cycle to cycle;**Absolute jitter https://www.fon.hum.uva.nl/praat/manual/PointProcess__Get_jitter__local__absolute____.html (accessed on 10 May 2021)**: it is the average absolute difference between consecutive periods in seconds;**RAP jitter https://www.fon.hum.uva.nl/praat/manual/Voice_2__Jitter.html (accessed on 10 May 2021)**: Relative Average Perturbation, the average absolute difference between a period and the average of it and its two neighbours, divided by the average period;**Local shimmer**: amplitude variation of the sound wave [25];**F1 mean**: it is the first formant https://www.fon.hum.uva.nl/praat/manual/Formant__Track___.html (accessed on 10 May 2021);**F2 mean**: it is the second formant;**F3 mean**: it is the third formant;**F4 mean**: it is the fourth formant;**Formant dispersion**: it is the difference between F4 and F1 divided by 3;**Mean intensity https://www.fon.hum.uva.nl/praat/manual/Intensity__Get_mean___.html (accessed on 10 May 2021)**: the mean (in dB) of the intensity values of the frames within a specified time domain;**Speech rate**: number of syllables divided by file duration in seconds;**Signal to Noise ratio (SNR)**;**LTAS https://www.fon.hum.uva.nl/praat/manual/Spectrum__To_Ltas__1-to-1_.html (accessed on 10 May 2021)**: it is the mean of logarithmic power spectral density as a function of frequency, computed over the entire domain frequency (from 0 Hz to 5000 Hz);**LTAS slope**;**LTAS standard deviation**;**RMS energy**: root-mean-square of energy;**Spectrum centre of gravity (SCG) https://www.fon.hum.uva.nl/praat/manual/Spectrum__Get_centre_of_gravity___.html (accessed on 10 May 2021)**: it is the average of frequency over the entire spectrum, weighted by the power spectrum;**Spectrum standard deviation https://www.fon.hum.uva.nl/praat/manual/Spectrum__Get_standard_deviation___.html (accessed on 10 May 2021)**: it is the variance of the frequencies in the spectrum;**Band Energy difference https://www.fon.hum.uva.nl/rob/NKI_TEVA/TEVA/HTML/Analysis.html (accessed on 10 May 2021)**: it is the ratio between the average power over low (between 0 Hz and 500 Hz) and high (between 500 Hz and 4000 Hz) frequency bins in decibel scale;

For this experiment, we decided to exclude speaker 314 because, from the Shift size point of view, its distance from the mean of all speakers’ OR is equal to 3.79 standard deviations, so for the Z-score rule [26] it could be treated as an outlier point. Thus, for the second experiment, we take into account just 29 speakers.

To generate the speaker’s voice features vector, we used an automated voice analysis software named Voice Lab [27]. Voice Lab is written in Python and relies heavily on a package called parselmouth-praat, which essentially turns Praat’s [28] source code written in C and C++ into a Python interface. Voice Lab computes all of the analysis parameters, but the user can always use his own. These are the same parameters as in Praat, and they do the exact same thing because it is Praat’s source code powering everything. All of the code is open-source and available on our GitHub repository https://voice-lab.github.io/VoiceLab (accessed on 10 May 2021). This experiment uses just IDEA speakers.

## 4. Analysis of Experimental Results

In this section, we will report all the results of both experiments. Specifically, Section 4.1 shows the first experiment results with speakers of IDEA, while Section 4.2 with speakers belonging to CLIPS database. Then, in Section 4.3 the results of the second experiment over just IDEA speakers are shown.

### 4.1. First Experiment Results—IDEA

The first experiment’s results show that, for all the 30 speakers with dysarthria, there exist an OR where the ASR performance is optimised. Table 2 shows the OR vectors in terms of the mean and standard deviation of window and shift values, for each subject. The average and standard deviation of the WER obtained from the ASR trained using the corresponding OR are shown as well. Furthermore, the WERs obtained using state-of-the-art values of window size and time shift, 25 and 10 ms respectively are reported and indicated as baseline values. The results in Table 2 are sorted according to the average improvement achieved by using OR values with respect to baseline/state-of-the-art values. Specifically, the first eight subjects do not have shown any improvement by using the OR. For example, for speakers 215, 305, 211, and 213 the mean of the WER of an ASR system that uses a window and shift sizes within the OR, is greater than the baseline one.

For speakers 401, 311, 206, and 323 the WERs assume similar values.

On the other hand, starting from speaker 301 until the 306 of the Table 2, using OR instead of baseline can improve the ASR performance up to 58%.

Figure 3 show the WER curves depending on window and shift sizes for the speakers 215, 306, 323, and 404. The speaker’s OR is located in the dark blue zone, where the WER is minimised. It is interesting that the shift value affects the WER value much more than the window one. This could be inferred also from the window standard deviation values, which are higher than the shift ones. This behaviour confirms the finding of [13].

### 4.2. First Experiment Results—CLIPS

The first experiment results for CLIPS speakers show that for unimpaired speech, there also exists an OR where the WER is minimised, and the OR shift size is very similar to the state-of-the-art value. Table 3 shows the OR for all 10 CLIPS speakers (5 females and 5 males) and the mean and standard deviation of the WER of all the points within the OR. The WER values are also compared with the baseline one.

The WER values do not show relevant differences with respect to state-of-the-art/ baseline values.

### 4.3. Second Experiment Results—IDEA

Table 4 shows the linear correlation coefficients with uncorrected *p*-values among window and shift mean values of the OR vector and the speaker’s voice features vectors. The highlighted parts are those that are significant after a BY correction, by limiting the false discovery rate equal to 0.1.

Regarding window size, there exists a significant (*p* = 0.0025) correlation coefficient equal to 0.5398 with the Spectrum Center of Gravity. Nonetheless, this finding did not survive the BY correction (adjusted *p* = 0.22).

From the shift point of view, which is the most important parameter as said in Section 4.1, the most important features are Local jitter, Absolute jitter, RAP jitter, and LTAS mean.

## 5. Discussion

From a technological point of view, in this work we exploit GMM-HMM architecture for the ASR design. As stated in the introduction, several works highlight the advantages of DNN-HMM architecture with respect to GMM-HMM one. We have to stress that this is the case when a large amount of data is available to train the DNN model. In the case of small training sample sizes, the GMM model was shown to have better performance [16].

Table 2 reports the results of the second experiment over IDEA database. As said in Section 4.1, the first eight subjects do not have shown any improvement by using the OR. Despite this, the WER values are significantly low, thus we can assume that the performances are comparable to each other. For speakers 401, 311, 206, and 323 the WERs assume similar values. Therefore, for these speakers, using the OR instead of the baseline is not relevant. For the other speakers, there exists an improvement in the ASR performance by using the OR parameters. These improvements become much more significant in the speakers in whom the baseline has worse performance. For instance, for speaker 314 the baseline WER is 80.67%, which means that the ASR system is useless. Using the OR instead, the WER goes to 43.15% that makes the ASR system much more usable from the dysarthric user. The same reasoning can be applied to speakers 405, 402, 307, 321, and 306.

From Table 3, as in the case of the speakers with dysarthria, we can infer that the window size is not so important for the ASR performance. This is because the WER values do not show relevant differences with respect to state-of-the-art/baseline values. This is also due to the large standard deviation of the window size parameter, thus indicating that this parameter can be freely chosen within a large interval without affecting optimal performances. This could be inferred also from the Figure 4. For the shift size parameter, the mean value tends to be similar to the baseline one and the standard deviation is very tiny. This result confirms that the baseline shift size is a good value for unimpaired speech. From the WER values comparison point of view, using OR parameters instead of baseline for unimpaired speech is not so worthy.

From the second experiment, regarding window size, the correlation analysis showed high correlation coefficients with some speech features related to speech spectral content. Specifically, increasing window sizes were increasing at increasing levels of SCG, spectrum standard deviation, and BED. Increasing BED value means that the energy of low frequencies (from 0 Hz to 500 Hz) become more meaningful than high frequencies (above 500 Hz). On the other hand, the SCG value varies between 21,337 Hz and 75,729 Hz with an average of 46,099 Hz. So it is reasonable to say that the high correlation between BED and SCG with window size might indicate the great importance of the vocalised signal, typical of frequencies around 500 Hz, for the window size parameter.

Regarding window shift, correlations that are more significant were highlighted. Specifically, jitter-related measures seem to be the best feature candidates to predict the window shift. Specifically, at an increasing level of jitter, a decreasing window shift is required. Since jitter is related to cycle-to-cycle variations of the fundamental period thus representing the deviation from the periodicity of the speech signal. Thus, it is interesting that window shift is negatively correlated with such a measure. We have to point out that this measure is estimated from words, so it is not so straightforward to relate it to its original meaning, which is more correctly obtained from sustained vowels. However, when this measure is applied to words we were still getting interesting information about voice quality. In this scenario, a higher level of perturbations of fundamental frequency implies a shorter time window time shift and thus finer temporal information. The obtained results show a significant correlation with the speech spectral information contained in the LTAS. Specifically, an increase of speech power between 20 Hz and 5000 Hz is correlated with an increased window time shift. This might be related to the fact that an increase of speech power might be related to a better speech production quality, from a sound intensity point of view. In this case, coarser information is needed. This seems to be confirmed by the correlation values obtained by RMS energy and Mean Intensity features that show significant or close-to-significance *p*-values.

## 6. Conclusions

In this work, we conducted two experiments regarding speech analysis of Italian speakers with and without dysarthria.

The first experiment extends the work done in [13] by adding more data. Mainly, we decided to use the voices of 30 speakers with dysarthria taken from the IDEA database [15], and the voices of 10 unimpaired speakers taken from the CLIPS database.

The aim of the first experiment was to validate the existence of an Optimal Region in the field of window and shift sizes, where the performance (measured in WER) of a speaker-dependent ASR system is minimised for a specific speaker. Specifically, we were interested to analyse how much the OR mismatch the baseline values for speakers with dysarthria. The results of the first experiment confirm the finding of [13], so there exists an OR for each speaker. In general, the WER is sensible of the shift size, while the window is not so important to optimise ASR performance. For some speakers, especially for those who have high WER, using OR parameters instead of baseline ones can increase the performance up to 58%. The unimpaired speaker ORs match the shift baseline value, while for the speakers with dysarthria the shift value may be different even far from the shift baseline value.

The aim of the second experiment was to find out if there is some correlation between some speaker’s voice features and their OR. This could be useful to locate the OR starting from few speaker recordings, avoiding recording a high number of speaker voice samples. There are selected 24 voice features, which were analysed in terms of correlation coefficients with OR mean values (window and shift). Spectral voice features showed a high correlation with the window size parameter. In particular, the features that emphasise the typical frequency components of the vocalised signal. On the other hand, jitter measures and LTAS information seems to be very significant to estimate the value of the best speaker window shift. 

## Figures and Tables

**Figure 1 sensors-21-06460-f001:**
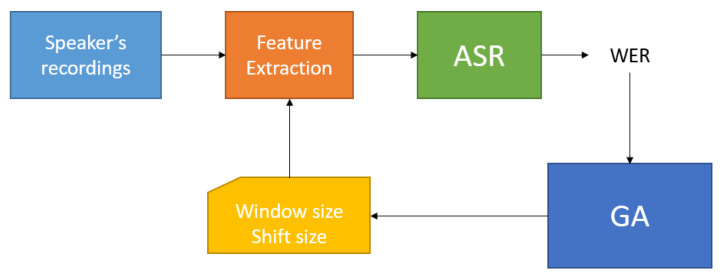
Block diagram of the experiment 1 working flow.

**Figure 2 sensors-21-06460-f002:**
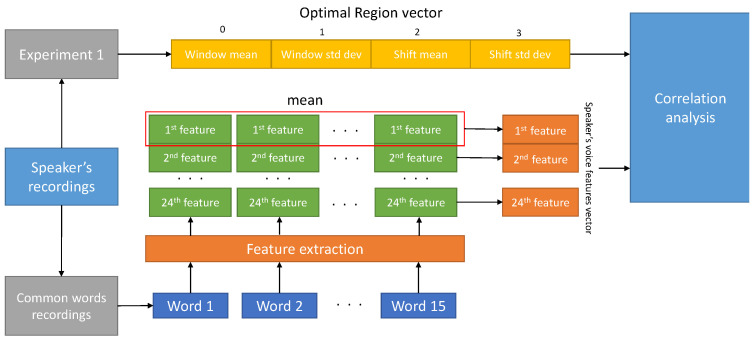
Block diagram of the experiment 2 working flow.

**Figure 3 sensors-21-06460-f003:**
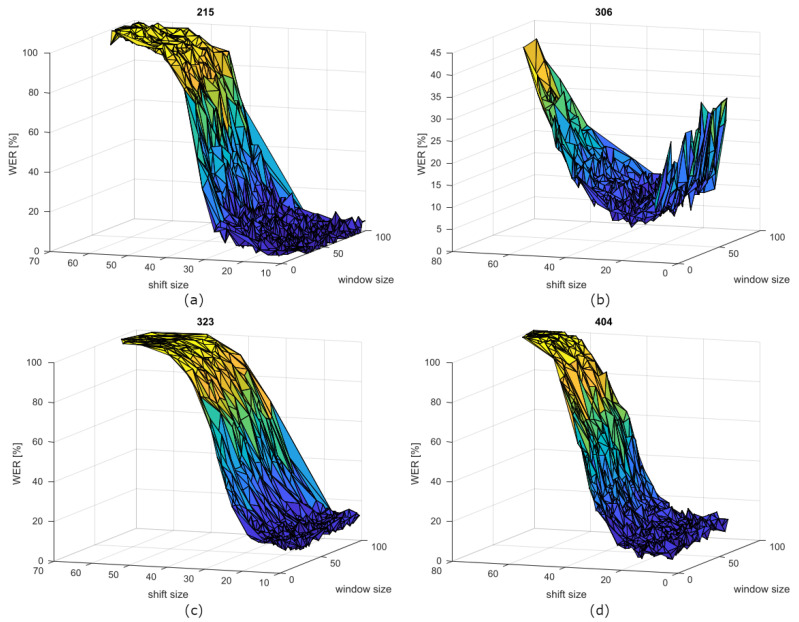
WER curves of speakers (**a**) 215, (**b**) 306, (**c**) 323, and (**d**) 404 depending on window and shift sizes.

**Figure 4 sensors-21-06460-f004:**
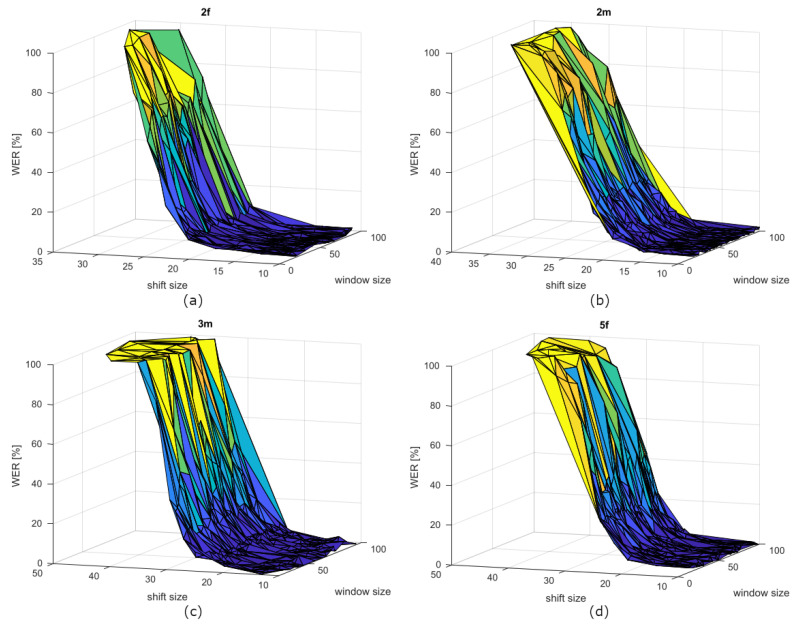
WER curves of speakers (**a**) 2f, (**b**) 2m, (**c**) 3m, and (**d**) 5f depending on window and shift sizes.

**Table 1 sensors-21-06460-t001:** Training and test sets speaker by speaker, sorted by the number of records contained in test set.

	Speaker’s ID	# Records	Train	Test
1	320	618	422	196
2	310	559	413	146
3	306	557	411	146
4	316	546	414	132
5	323	523	408	115
6	322	505	392	113
7	302	505	395	110
8	303	494	384	110
9	314	492	388	104
10	308	477	382	95
11	311	405	338	67
12	405	444	379	65
13	321	391	328	63
14	404	432	372	60
15	307	412	358	54
16	313	394	340	54
17	401	246	196	50
18	215	129	90	39
19	403	134	97	37
20	214	127	90	37
21	205	125	89	36
22	301	299	264	35
23	402	354	320	34
24	206	121	88	33
25	211	121	89	32
26	203	118	88	30
27	305	274	245	29
28	208	118	89	29
29	213	112	86	26
30	312	432	409	23
31	207	96	79	17
32	210	83	71	12
33	209	72	61	11
34	212	69	61	8
35	315	235	230	5
36	317	194	190	4
37	201	59	55	4
38	319	235	235	0
39	318	201	201	0
40	309	104	104	0
41	204	30	30	0
42	216	24	24	0
43	202	9	9	0
44	304	9	9	0
45	406	7	7	0

**Table 2 sensors-21-06460-t002:** Experiment results with IDEA database: speakers’ Optimal Regions and their WER compared with baseline (window and shift respectively 25 and 10 ms).

Spk ID	Window (ms)		Shift (ms)		WER (%)		
	Avg	Stdev	Avg	Stdev	Avg	Stdev	Baseline
215	57.34	21.17	16.07	3.75	2.68	3.05	0.768
305	48.84	20.27	20.32	4.13	9.05	4.29	6.9
211	45.14	19.84	13.92	2.07	7.49	5.37	5.94
213	62.28	23.49	15.55	3.87	12.49	5.76	11.54
401	54.30	19.73	18.45	4.37	3.98	3.25	3.8
311	44.77	16.74	13.64	2.48	8.71	3.20	8.36
206	56.00	19.41	18.79	5.20	2.69	3.28	2.73
323	51.58	20.68	13.84	2.20	10.10	2.32	10.61
301	52.74	25.78	13.56	2.55	16.66	5.15	18
308	45.35	14.29	14.15	2.08	6.41	3.27	7.37
316	47.98	19.30	15.43	2.20	8.52	2.56	9.93
322	60.81	19.90	21.34	4.35	12.03	2.66	14.87
208	57.39	19.51	21.63	4.85	4.10	3.36	5.17
404	46.48	17.92	16.89	3.64	5.94	2.60	7.5
214	40.09	17.04	11.21	1.25	7.14	5.20	9.19
313	46.08	21.54	17.65	2.62	11.53	3.78	15.37
312	56.37	22.79	18.51	3.83	2.79	3.03	3.92
205	49.39	22.05	15.30	2.70	3.35	2.73	4.72
310	53.25	17.28	19.77	3.02	1.87	1.27	2.67
303	41.39	13.44	18.82	2.36	4.50	1.71	6.54
320	65.85	17.02	24.17	4.94	7.26	1.36	10.87
302	49.09	17.58	18.44	3.03	2.60	2.14	4.55
203	60.09	19.57	18.51	5.51	6.35	5.78	11.66
403	64.95	22.06	27.78	10.79	1.91	2.21	3.51
314	66.87	16.37	38.80	8.73	43.15	6.14	80.67
405	48.88	17.63	17.15	2.26	8.59	4.12	16.92
402	42.60	20.51	20.13	2.03	7.54	5.57	16.18
307	49.78	14.74	15.68	3.22	6.40	3.37	13.89
321	43.57	16.84	18.49	3.02	6.87	3.24	16.35
306	55.42	17.04	27.25	4.91	7.89	1.97	19.11

**Table 3 sensors-21-06460-t003:** Experiment results with CLIPS database: speakers’ Optimal Regions and their WER compared with baseline (window and shift respectively 25 and 10 ms).

Spk ID	Window (ms)		Shift (ms)		WER (%)		
	Avg	Stdev	Avg	Stdev	Avg	Stdev	Baseline
1f	71.73	21.74	13.17	1.39	1.14	0.65	0.65
2f	56.91	21.57	13.26	1.65	1.02	0.52	0.68
3f	59.85	19.94	14.71	1.65	0.52	0.45	0.59
4f	60.87	21.61	11.74	0.89	1.46	1.02	0.68
5f	59.79	21.41	12.63	2.76	1.08	0.52	1.19
1m	70.40	21.41	10.54	1.22	1.33	0.81	1.11
2m	56.00	23.92	11.66	2.06	0.99	0.46	0.85
3m	64.73	22.95	13.37	2.14	1.23	1.05	0.94
4m	62.73	22.01	11.13	1.24	0.92	0.87	0.37
5m	52.88	23.67	12.44	1.86	1.02	0.69	0.74

**Table 4 sensors-21-06460-t004:** Linear correlation coefficients with uncorrected *p*-values among speaker’s voice features and Window and Shift mean values for IDEA speakers. The statistical significance after Benjamini–Yekuteli correction is shown in bold.

Features	Window Mean		Shift Mean	
	corr. coef.	*p*-Value	corr. coef.	*p*-Value
Duration in sec	0.1172	0.5448	0.1294	0.5034
Mean F0	0.1532	0.4276	−0.1092	0.5728
STD F0	−0.1056	0.5856	−0.374	0.0456
SHR	0.0256	0.8951	−0.2287	0.2327
Subharmonics pitch	−0.0805	0.678	0.0799	0.6803
Local jitter	−0.1857	0.3348	**−0.6993**	**0**
Absolute jitter	−0.2534	0.1847	**−0.6352**	**0.0002**
RAP jitter	−0.1794	0.3517	**−0.6516**	**0.0001**
Local shimmer	−0.2875	0.1305	−0.4762	0.009
F1 mean	−0.0968	0.6175	−0.0263	0.8923
F2 mean	−0.0007	0.997	0.0291	0.8807
F3 mean	0.1708	0.3756	0.0914	0.6373
F4 mean	0.1763	0.3604	0.0103	0.9578
Formant dispersion	0.3078	0.1043	0.0365	0.8509
Mean intensity	−0.0476	0.8063	0.4274	0.0207
Speech rate	0.1263	0.5138	−0.309	0.1029
SNR	0.2536	0.1843	0.3175	0.0932
LTAS mean	0.1306	0.4996	**0.5245**	**0.0035**
LTAS slope	0.1232	0.5242	0.1859	0.3342
LTAS std	−0.122	0.5284	0.2414	0.2071
RMS energy	−0.0116	0.9524	0.3516	0.0614
SCG	**0.5398**	**0.0025**	0.3867	0.0382
Spectrum std	0.4291	0.0202	0.0634	0.7437
Band Energy difference	0.443	0.0161	0.4837	0.0078

## Data Availability

Not applicable.

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
