# Peer review of "Optimising Speaker-Dependent Feature Extraction Parameters to Improve Automatic Speech Recognition Performance for People with Dysarthria"

_sensors, 2021, doi:10.3390/s21196460_

Round 1

Reviewer 1 Report

I enjoyed reading your paper. I think the topic is very relevant and useful for dysarthric speakers, their care partners, their clinicians and their community as a whole. I do have some comments/clarification questions, which I'm sharing below. I also recommend the authors seek out proof-reading services for English, as the manuscript requires extensive editing for language. I've done my best to provide a comprehensive review of edits:

Abstract:

In the presence of dysarthria, need to add article (the)

Instead of dysarthric speaker, please use speakers with dysarthria (people first language)

If there exists (-s is missing)

Same for ‘that minimizes’, the -s morpheme is missing

Same for ‘if a standard ASR system performs’, need to add -s for grammar (so 3rd person singular subject requires verb with -s)

In case of should be in cases of

His optimal parameters should be their optimal parameters

Introduction

We do not say dysarthric people but people with dysarthria

The following sentence requires clarification, as it seems that stuttering is the same as dysarthria, which would be inaccurate: 'Dysarthric people comprise individuals with a primary speech disorder such as stuttering'. Stuttering may occur in people with hypokinetic dysarthria but when it is the primary concern/speech symptom it may reflect neurogenic stuttering instead.

This sentence needs revision (despite the fact that…): Despite in the last years the Automatic Speech Recognition (ASR) systems have recorded an outstanding performance improvement

other kids of disability should be ‘other kinds’, revise for typo

in an everyday living is ungrammatical, you can say ‘in their daily lives’, instead

Please revise this sentence for grammar: An earlier study about acoustic and lexical model adaptation was investigated in [9] showing that average relative Word Error Rate (WER) reduction of 32 36.99% for an ASR system trained over a large vocabulary dysarthric speech database.

Exploiting a database sounds off, I recommend saying ‘using a database’ , for example

Materials:

A lot of effort has (not have) been spent

Correct for grammar: training and test sets results unbalanced as well.

What does ‘suitable for lexical coverage’ refer to?

The following sentence should be revised for style and grammar: Obviously, the test set contains recordings that do not appear in training one. I suggest deleting ‘obviously’, as it is too informal for an academic paper. The ending of the sentence is ungrammatical, so the authors need to address that.

I think the authors should introduce to the reader what they mean by training and tests sets for clarification purposes

The second experiment aims to finding possible correlations…aims at (not to)

What was the rationale for selecting the specified words in Experiment 2? Also, I’d like to read about the rationale for selecting the specified voice parameters

Figuring out in ‘Voice Lab figures out all of the analysis parameters’ is too informal, the authors should select more formal verbs

In the results section, what tests were used to compute statistical analysis?

This sentence needs to be revised for clarity: These improvements become much more significant in the speakers in whom the baseline performs worse (the baseline is not animate, so it cannot perform)

This sentence needs a subject: when applied to words was still getting interesting information about voice quality

Thank you

Author Response

Dear reviewer,

thank you for your revisions and corrections.

All the language editing suggestions have been addressed and highlighted in the final version.

Furthermore, other reviewer questions are addressed below:

  • What does ‘suitable for lexical coverage’ refer to?
    LP sentences list of CLIPS database has been developed from the CLIPS owner in order to cover all the Italian phonemes that do not appear in LB sentences list.
  • I think the authors should introduce to the reader what they mean by training and tests sets for clarification purposes:
    We added a sentence at line 119 with this purpose
  • What was the rationale for selecting the specified words in Experiment 2? Also, I’d like to read about the rationale for selecting the specified voice parameters
    As explain at lines 213-214, the selected words are the only ones in common between all patients. That is, after the annotation process, they are the only ones that have at least one record among all patients involved.
    For the voice parameters, as explained at lines 239-241, we used those that are usually used in voice assessment.

Best regards.

Reviewer 2 Report

Line 142/143:  The mention of GA as an optimizer because it is parallelizable seems distracting… if the goal was parallelizability, why not choose something more straightforward - standard hyperparameter optimization techniques such as grid search or binary search?  Ray is a common tuning mechanism for machine learning hyperparameters which can be parallelized across multiple computing resources:  https://docs.ray.io/en/latest/tune/index.html

If the GA is important, it should be completely explained (perhaps in an appendix), to include exactly what the chromosome is, how it is represented (a pair of numbers?  a set of bits representing those floating point numbers?) How do mutation and recombination work in this GA – especially if there are only 2 members of the chromosome.  What about the GA pool size, generation characteristics, and other details?  Is your fitness function the WER?  be explicit.  What are the GA’s “convergence conditions” which allow you to decide convergence?    How do you ensure diversity of exploration and avoid genetic stagnation?

Algorithm 1, line 3:  the “While” is unnecessary as this loop is a DO UNTIL loop.  “Do until GA convergence”

Line 207:  The feature “Duration in seconds” seems suspicious and hard to implement.  In real-world speech recognition tasks, the audio is a continuous stream… shouldn’t your model NOT have this information a-priori?

Table 2 is difficult to read and interpret.  What is the message - that every speaker is different?  That they are actually similar?   The format of presenting every speaker’s window and shift and WER makes it difficult to interpret your message.  Figure out what you want to say about this information and then rework the presentation appropriately… for example, if you want to show that only certain speakers had certain outcomes (benefit from optimal region determination) then HIGHLIGHT (bold) those speakers.   Alternately if you are trying to show the DISTRIBUTION of outcomes, then use a histogram.

Line 301/302:  the discussion about GA being implemented previously (in other work) in java and now (in this work) in python is distracting and perhaps worrisome.   What is your purpose for explaining this in the discussion instead of the methodology?  Are you concerned that the outcomes/convergence properties of the two different GA approaches might be different simply due to which language they were implemented in?  If so, you need to provide some evidence to the reader that you have investigated this possibility and ruled it out.  However, if your goal is to suggest that your work is novel because you used a different language for your GA then you are mistaken… this is not a contribution and should be removed from the discussion – describe the implementation language of the GA earlier in the methodology section when you describe all of the other characteristics of the GA I mentioned in the previous comment.

Lines 355-378: Your conclusions section has no Future Work… why not?

Here are some Suggestions for Future Work section: 

1) Repeat this experiment using continuous speech which YOUR system has to segment as part of ASR. (your model should NOT be allowed to have small clips of single word pronunciations where you know the whole word is contained in the clip, and the “duration” of the clips should not be part of the feature set. )

2) Repeat this experiment with a more common language such as English to see if the findings hold across languages, or if they are specific to Italian.

Writing Style:

Overuse of acronyms makes the paper harder to read then necessary.  Suggest spelling out “Optimal Region” and “Speaker-Dependent” on EVERY location they are used in the document… overusing OR and SD makes it hard to sift through the analysis and results sections.

English language proofreading:  in several locations, articles (“the”) are present and should be removed.  For example, “The Figure 1 shows…” should be “Figure 1 shows”

Author Response

Dear reviewer,

thank you for your revisions and comments.

All the reviewer questions are addressed below:

  • The mention of GA as an optimizer because it is parallelizable seems distracting… if the goal was parallelizability, why not choose something more straightforward - standard hyperparameter optimization techniques such as grid search or binary search?  Ray is a common tuning mechanism for machine learning hyperparameters which can be parallelized across multiple computing resources:  https://docs.ray.io/en/latest/tune/index.html:
    The parallelizability is not the goal of using GA but is one of the interesting feature of this algorithm. Probably the sentence is misleading so we decided to change it by removing the "parallel" part (see line 162).
  • If the GA is important, it should be completely explained (perhaps in an appendix), to include exactly what the chromosome is, how it is represented (a pair of numbers?  a set of bits representing those floating point numbers?) How do mutation and recombination work in this GA – especially if there are only 2 members of the chromosome.  What about the GA pool size, generation characteristics, and other details?  Is your fitness function the WER?  be explicit.  What are the GA’s “convergence conditions” which allow you to decide convergence?    How do you ensure diversity of exploration and avoid genetic stagnation?
    We added a paragraph in Section 2.4 where we explain all the GA parameters and process in details (see lines 163 - 182).
  • Algorithm 1, line 3:  the “While” is unnecessary as this loop is a DO UNTIL loop.  “Do until GA convergence”
    The Algorithm 1 has been changed as asked by the reviewer
  • The feature “Duration in seconds” seems suspicious and hard to implement.  In real-world speech recognition tasks, the audio is a continuous stream… shouldn’t your model NOT have this information a-priori?
    In our real scenario, we will have audio files already segmented for each recorded word. This is because a word may be pronounced at different times from subject to subject (very disarticulated subjects are likely to take longer to pronounce a word). 
  • Table 2 is difficult to read and interpret.  What is the message - that every speaker is different?  That they are actually similar?   The format of presenting every speaker’s window and shift and WER makes it difficult to interpret your message.  Figure out what you want to say about this information and then rework the presentation appropriately… for example, if you want to show that only certain speakers had certain outcomes (benefit from optimal region determination) then HIGHLIGHT (bold) those speakers.   Alternately if you are trying to show the DISTRIBUTION of outcomes, then use a histogram.
    The message of table 2 is to show the optimal regions of all the subjects and the relative performances compared with the baseline. 
    From this information we can deduce that it is possible to subdivide the subjects according to how much the WER of the new technique differs from the baseline.
    In particular, we have four groups: the first group contains subjects whose WER values are worse than the baseline; the second group has subjects whose WER values are very similar or equal to the baseline; the third group contains subjects whose WER values are moderately better than the baseline; the subjects in the fourth group have WER values that are much better than the baseline.
  • the discussion about GA being implemented previously (in other work) in java and now (in this work) in python is distracting and perhaps worrisome.   What is your purpose for explaining this in the discussion instead of the methodology?  Are you concerned that the outcomes/convergence properties of the two different GA approaches might be different simply due to which language they were implemented in?  If so, you need to provide some evidence to the reader that you have investigated this possibility and ruled it out.  However, if your goal is to suggest that your work is novel because you used a different language for your GA then you are mistaken… this is not a contribution and should be removed from the discussion – describe the implementation language of the GA earlier in the methodology section when you describe all of the other characteristics of the GA I mentioned in the previous comment.
    We agree with the reviewer and we decided to remove the part that mentions earlier implementation and add more details about GA implementation.
  • Your conclusions section has no Future Work… why not?
    We added a Future Work part in Conclusion section as suggested by reviewer.